# Fatty Acid Profiles from Routine Milk Recording as a Decision Tool for Body Weight Change of Dairy Cows after Calving

**DOI:** 10.3390/ani10111958

**Published:** 2020-10-23

**Authors:** Franziska Dettmann, Daniel Warner, Bart Buitenhuis, Morten Kargo, Anne Mette Hostrup Kjeldsen, Niels Henning Nielsen, Daniel M. Lefebvre, Debora E. Santschi

**Affiliations:** 1Lactanet, Sainte-Anne-de-Bellevue, QC H9X3R4, Canada; f.dettmann@lkv-we.de (F.D.); dwarner@lactanet.ca (D.W.); dlefebvre@lactanet.ca (D.M.L.); 2LKV Niedersachsen e.V., 26789 Leer, Germany; 3Center for Quantitative Genetics and Genomics, Aarhus University, 8830 Tjele, Denmark; bart.buitenhuis@qgg.au.dk (B.B.); morten.kargo@qgg.au.dk (M.K.); 4SEGES, 8200 Aarhus N, Denmark; amk@seges.dk; 5RYK, 8200 Aarhus N, Denmark; nhn@ryk-fonden.dk

**Keywords:** decision support, Fourier-transform infrared, dairy herd improvement, bodyweight loss, machine learning

## Abstract

**Simple Summary:**

Dairy cows mobilize their body reserves to maintain milk production after parturition. If excessive, this may be detrimental for their health and reproduction performance. Detecting cows with an important bodyweight loss early on is essential to treat or manage these cows accordingly. Routine milk recording samples were used to develop a prediction model for early lactation body reserve mobilization in commercial dairy farms using a machine-learning approach. Bodyweight loss was mainly explained by decreased short-chain fatty acids and increased C18:0 fatty acids in milk. An early warning system for cows with an important bodyweight loss may be implemented routinely.

**Abstract:**

Cows mobilize body reserves during early lactation, which is reflected in the milk fatty acid (FA) profile. Milk FA can be routinely predicted by Fourier-transform infrared (FTIR) spectroscopy, and be, thus, used to develop an early indicator for bodyweight change (BWC) in early lactating cows in commercial dairy farms. Cow records from 165 herds in Denmark between 2015 and 2017 were used with bodyweight (BW) records at each milking from floor scales in automatic milking systems. Milk FA in monthly test-day samples was predicted by FTIR. Predictions of BWC were based on a random forest model and included parity, stage of lactation, and test day milk production and components (fat, protein, and FA). Bodyweight loss was mainly explained by decreased short-chain FA (C4:0–C10:0) and increased C18:0 FA. The root mean square error (RMSE) of prediction after cross-validation was 1.79 g/kg of BW (R^2^ of 0.94). Model evaluation with previously unseen BWC records resulted in reduced prediction performance (RMSE of 2.33 g/kg of BW; R^2^ of 0.31). An early warning system may be implemented for cows with a large BW loss during early lactation based on milk FA profiles, but model performance should be improved, ideally by using the full FTIR milk spectra.

## 1. Introduction

In early lactation, most dairy cows face energy deficits, due to an imbalance between feed intake and milk production requirements. Negative energy balance varies between cows in duration and extent [1], but postpartum cows typically mobilize body reserves [2,3] to maintain milk fat production [4]. Due to mobilization of body fat reserves, a certain loss in bodyweight (BW) is to be expected during early lactation, and a post-calving deterioration of 0.5 to 1.0 body condition scores (BCS) units on the 5-point scale are generally accepted in practice [5]. However, excessive mobilization of body fat might increase metabolic disorders and reproductive issues [3].

Mobilization of body fat is also reflected in a modified milk fatty acid (FA) profile. De novo FA synthesis in the mammary gland is decreased in early lactation, whereas greater proportions of preformed FA from body fat mobilization are incorporated into milk [6,7]. While BW is not typically routinely recorded in dairy farms, milk composition, and milk FA profiles can be routinely determined by Fourier-transform infrared (FTIR) spectroscopy. Therefore, this rapid analysis offers an opportunity to develop a regular, early indicator for bodyweight change (BWC) based on the FA profile of milk recording samples. The objective of this study was to evaluate whether milk FA by FTIR analysis can be used to predict BWC in early lactating cows from milk recording samples collected in commercial dairy farms. 

## 2. Materials and Methods 

Cow information, test day milk yields, and milk components, and BW records of individual cows were obtained by the Danish Cattle database (SEGES, Skejby, Denmark). All herds were equipped with an automatic milking system (AMS; Lely Nordic A/S, Fredricia, Denmark). No animal handling was required, and all producers gave their informed consent for the use of their data. 

### 2.1. Test Day Records for Milk Components and Fatty Acids

The initial dataset included 197,058 test day records of 34,870 Danish Holstein cows from 169 herds in Denmark, collected from March 2015 to March 2017. Cows were between 5 and 305 days in milk (DIM) with parity 1 to 3. Test day records comprised 17 variables: Test day milk yield; fat; protein; somatic cell count; beta-hydroxybutyrate (BHB); the individual milk FA myristic (C14:0), palmitic (C16:0), stearic (C18:0) and oleic (C18:1) acid; groups of milk FA according to their degree of saturation (saturated FA, SFA; mono-unsaturated FA, MUFA; and poly-unsaturated FA, PUFA), and their chain length (short-chain FA, SCFA; medium-chain FA, MCFA; and long-chain FA, LCFA). Milk samples were analyzed as regular milk recording samples within a dairy herd improvement (DHI) service, using FTIR spectroscopy on Foss MilkoScan FT+ and FT 6000 (Foss Electric A/S, Hillerød, Denmark), equipped with special software (Foss Application Note 64; Foss, Hillerød, Denmark) for prediction of aforementioned FA. 

Following procedures previously described [8], observations with one or more missing FA fractions and abnormal observations were removed, defined as having a PUFA concentration greater or equal to the MUFA concentration; a ratio of the sum of SFA, MUFA, and PUFA contents to total fat content of less than 0.825 or greater than 1.075 (values were chosen such that 5% of remaining observations were removed); and a ratio of the sum of SCFA, MCFA, and LCFA contents to the sum of the SFA, MUFA, and PUFA contents of less than 0.84 or greater than 1.04 (values were chosen such that 1% of remaining observations were removed).

Test day records between 5 and 35 DIM were retained for the model to predict BWC in early lactation. As test day records were generally collected monthly, for most cows (87%), only the first test day during early lactation was retained. The remainder was tested twice during early lactation.

### 2.2. Bodyweight Records

Bodyweight data were recorded by floor scales in Lely AMS (Lely Astronaut A3 and A4; Lely Nordic A/S, Fredricia, Denmark) at each milking. Floor scales were installed at the bottom of the AMS and were composed of four load cells in each corner. The expected tolerance for each load cell was ±10% based on the recommendations by Lely (Lely Nordic A/S, Fredericia, Denmark). The participating producers were expected to calibrate the floor scales regularly in line with recommendations by Lely. However, due to the nature of the study using large scale farm records, it was not possible to ascertain whether calibrations were done. 

The initial dataset included 28,581,762 BW records of 35,787 Danish Holstein cows from 168 herds in Denmark, collected from January 2015 to September 2017. Cows were between 5 to 305 DIM with parity 1 to 3. Individual BW was corrected for potential discrepancies between AMS within a herd for herds equipped with more than one AMS, and smoothed using locally weighted smoothing procedures to correct for daily differences in BW (Figure 1), following procedures described elsewhere [9]. Bodyweight records were removed if the BW was less than 300 kg or greater than 1100 kg, and if the BW was more than 3 × the standard error of the smoothed BW. Smoothed BW records were then transformed to a relative daily BW change per DIM for each cow (expressed as g BWC/kg of BW), with the aim to generate a standardized metric across cows with different BW, as follows:BWC_DIM(*x*)_ = (BW_DIM(*x*)_ − BW_DIM(*x* − 1)_)/BW_DIM(*x* − 1)_(1)
where, BWC_DIM(*x*)_ is the daily BW change (g/kg of BW) at DIM *x*; BW_DIM(*x*)_ is the smoothed daily BW (kg) at DIM *x*; and BW_DIM(*x* − 1)_ is the smoothed BW (kg) of the previous day. 

### 2.3. Data Editing

Records of BWC were merged with FA profiles of the respective test date. As milk fatty acid profiles were shown to change mostly within the first two days up to four days following a dietary transition [10,11], it was assumed that the FA profiles at test date were the results of the BWC observed during the previous three days. An average BWC was therefore calculated over the three days preceding the test date record (i.e., BWC from day 3 throughout test day) and merged with the respective test day record. As the average daily BCW for DIM 5 and 6 could not be computed, the earliest test day records retained were those for DIM 7. The final data set included 19,138 test day records between DIM 7 and 35 with the respective average BWC. In total, 16,847 early lactation Danish Holstein cows across 165 herds were retained for analysis (Table 1). Cows had an average milk yield of 35.9 kg at test day (standard deviation (SD) of 10.90; range from 2.1 to 76.8 kg).

### 2.4. Prediction Model

Feature selection was conducted through a principal component analysis with the aim of keeping only the most important information in the dataset. All variables included in the first three principal components (explaining 80% of the total variance) after plotting eigenvalues according to their size were retained [12]. Somatic cell count was omitted from the final dataset, due to its low contribution (<5%) and low correlation (r < 0.50) with any of the three principal components. The final dataset comprised 16 variables, namely, the milk components and milk FA presented in Table 1, milk yield, parity (1, 2, and 3), DIM, test date year, and month. None of the predictors had a near-zero variance. Variables were normalized to a mean of 0 and a standard deviation of 1. 

Prediction of BWC at test day was based on decision tree induction using a random forest algorithm [13]. A random forest is an ensemble of multiple decision trees, and thus, essentially the sum of piecewise linear functions. The dataset is split into smaller sets that become more manageable by repeatedly selecting a random sample with replacement from the training set and fitting multiple decision trees to these samples (i.e., each decision tree is shown a different training set). In addition, predictors are randomly selected to prevent trees from becoming correlated. Thus, a random forest model may deal with a multitude of linear and nonlinear relationships among predictors, and the complexity inherent to a high-dimensional dataset [14]. 

### 2.5. Model Development

The model was implemented in R (version 3.5.0; R Foundation for Statistical Computing, Vienna, Austria) using the modeling package workflow caret [15]. The dataset was randomly split into a training set (80%) used for model preparation, and an evaluation set (20%) containing output values withheld from the algorithm with the aim to evaluate predictions from the trained model with previously unseen BWC records. Data splitting was done based on two different stratification approaches to ensure that data of a specific cow (grouped by animal) or specific herd (grouped by herd) were used exclusively either to train or to validate the model. The aim of the latter approach was to ensure that herd-level information was not inadvertently passed to both the training and evaluation set when cows from the same herd were assigned. 

The model was prepared using 10-fold cross-validation with three iterations. For each iteration, a model was trained on nine splits of the data set and cross-validated on the remaining part of the data set (i.e., one split). Like the data splitting approach, described above, either cows or herds were randomly assigned to the cross-validation folds—such that data of a specific cow or herd are used exclusively either to train or to cross-validate the model. The optimal hyper-parameter configuration for the random forest algorithm was evaluated based on the repeated cross-validation through a random parameter search [16] of the number of classification trees (500, 1000, 1500, and 2000 trees) and the number of randomly selected predictors (1 to 10). The final model configuration was 500 trees with three randomly selected predictors. Model accuracy was estimated on the average of the 10-fold repeated cross-validation and compared to the validation set based on Pearson’s r correlation coefficient, Lin’s concordance correlation coefficient (CCC), the coefficient of determination (R^2^), the root mean squared error of prediction (RMSEP), the mean absolute error (MAE), the ratio of performance to deviation (RPD; defined as the SD of observed values divided by the RMSEP [17]), and the ratio of performance to the interquartile range (RPIQ; defined as the interquartile range of the observed values divided by the RMSEP [18]). The predictive performance of the model is best with small values for RMSEP and MAE, and large values for r, CCC, R^2^, RDP, and RPIQ. The variables contributing most to the model development (i.e., the global variable importance scores) were extracted, and overall variable importance was determined for each tree based on the mean decrease in node impurity as determined by the Gini index and averaged across trees. The model response was visualized with an alluvial diagram for the variables with the highest feature importance with variables normalized to a mean of 0 and a standard deviation of 1. In addition, localized variable importance scores were extracted for randomly selected cows to interpret the model response for individual cows by using the break-down approach implemented in the DALEX package in R 3.5.0 [19]. This procedure sequentially attributes variables to the model output, thus, decomposing predictions and interpreting the link between input variables and model output [20].

## 3. Results and Discussion

### 3.1. Milk Fatty Acid Profiles

Milk FA profiles differed with parity and lactation stage, as shown in Figure 2. In early lactation (defined as 7 to 35 DIM in the present study), the concentrations of SCFA and MCFA increased, whereas the initially high levels of LCFA decreased rapidly as DIM progressed. These changes in the milk FA profiles can be explained by increased body fat mobilization [2,3] to maintain milk production [4]. With increasing parity, the concentration of FA increased, due to a greater milk fat production and a stronger body fat mobilization, due to a stronger negative energy balance in cows at later parity [21]. Our findings are in line with those reported in the literature. Molar proportions of the SCFA C10:0 to C14:0 were significantly lower in early lactation than in mid-lactation [7]. De novo synthesis of SCFA in the mammary gland might be thereby inhibited by LCFA from body fat [6] and linked to the adaptation of the rumen microflora to the lactation diet. 

In adipose tissue, the LCFA C18:1c9 and C18:0 and the MCFA C16:0 account for nearly 90% of the FA at approximately equal molar proportions [22]. With the mobilization of body reserves during early lactation, these FA would be more likely incorporated into milk fat. Nonetheless, while daily yields of C18:0 and C18:1 were indeed greater in early lactation, daily yields of C16:0 were smaller in early lactation in the present study. These results confirmed previous findings [23,24].

### 3.2. Relative BW Change

Bodyweight curves were distinctive across parities, with a lower average BW for cows in first parity (603 ± 66.1 kg; mean ± SD) than cows in second (656 ± 67.7 kg), or third parity (689 ± 70.2 kg). The relative daily BWC in early lactation was −0.52 ± 2.65 g/kg of BW, −0.64 ± 2.82 g/kg of BW, and −0.82 ± 5.53 g/kg of BW for cows in first, second and third parity, respectively (Figure 3). These results are comparable to the range of BW losses reported for different parities [21] with average BW losses of 6.5%, 8.5%, and 8.4% from calving to nadir BW for cows in first, second, and third or greater parity, respectively. Overall, these results suggest that the level and duration of negative energy balance increased with later parity in line with findings in the literature [21]. However, these may not be valid for all cows as cows with a high BW loss in the previous lactation were shown to have increased BW losses at later parity (5.87%, 6.49%, and 7.89% BW loss for cows in first, second and third parity, respectively, from calving until five weeks of lactation), but cows with a low BW loss in the previous lactation had decreased BW losses at later parity (7.27%, 4.83%, and 5.45% BW loss) [25]. Variation in BWC may be also be explained by differences in gastrointestinal size and content [26]. In the present study, BWC reflects variation in BW as mobilization or deposition and changes in gastrointestinal fill [3].

In practice, the extent of BW loss might be more easily interpreted through a deterioration of the body condition. While a BCS loss of no more than 0.5 to 1.0 units on a 5-point scale should be targeted in practice [5], an average daily BW loss of 3 g/kg of BW over a 28-day period from DIM 7 to 35 can be translated into a theoretical cumulative loss of 1.2 to 1.5 BCS on the 5-point scale whilst considering the 10% range of error of the floor scales (an average increase of 41 kg BW per unit of increase in BCS for Holstein and mixed herds was assumed [27]).

### 3.3. Model Performance and Evaluation

Daily BWC during early lactation was predicted from test-day milk components and FA using decision-tree induction through a random forest algorithm. Decision-tree induction is a flexible approach using all available variables. However, the variables that most influenced the predictions with an overall importance of over 70% across all decision trees are SCFA, C18:0, and MUFA (Figure 4). Due to the presence of collinearity among the individual FA, care should be taken as to the ranking of the individual variable as their importance might not be well defined. Milk BHB was not considered an important feature in the present study despite its known association with hyperketonemia, a metabolic disorder commonly affecting cows in negative energy balance after parturition [28,29]. However, the prevalence of 8.4% for elevated milk BHB (>0.15 mmol/L) in the present study was considerably lower than that observed in dairy cows in Canada (21% and 22.6%) [28,29], and across Europe (21.8%) [30]. In addition to possible genetic differences, the low BHB prevalence might explain its low overall importance for BWC in our model.

Using a model-agnostic approach, the model predictions can be investigated in more detail (Figure 5). For cows with a large BW loss in early lactation, the predicted BWC were explained by generally lower concentrations of SCFA and higher concentrations of C18:0 and MUFA. In contrast, for cows with a BW gain in early lactation, predicted BWC was explained by generally higher concentrations of SCFA and lower concentrations of C18:0 and MUFA. Decision rules may thereby vary for each prediction instance using decision-tree induction. For instance, the predicted BWC was influenced by different predictors for two randomly selected cows from the same herd with a relatively large BW loss in early lactation (Figure 6). For a cow at DIM 25 with a predicted BWC of −3.69 g/kg of BW (observed BWC of −4.05), BWC was mainly positively influenced by LCFA and negatively influenced by SCFA and C18:0 (Figure 6A). In contrast, for a cow at DIM 10 with a predicted BWC of −1.98 g/kg of BW (observed BWC of −2.22), BWC was mainly negatively influenced by DIM, C14:0, but positively influenced by LCFA and milk BHB (Figure 6B). This example further shows that decision rules can be interpreted for single prediction instances despite the complexity of a random forest algorithm relying on the prediction of a multitude of decision trees. By integrating additional information, such as diet information, to test day milk fatty acids, it might be possible to investigate the specific pathways shown below in more detail.

Predicted BWC after cross-validation (median ± absolute deviation; −0.44 ± 2.25 g/kg of BW per day) did not significantly differ (*p* > 0.05) from observed BWC (−0.41 ± 1.64 g/kg of BW per day), suggesting that a random forest algorithm was capable of predicting BWC. Nonetheless, predictions of BWC were off the range of the observed BWC (interquartile range of 3.04 for predicted values as compared to 2.24 g/kg of BW for observed values). The prediction model tended to underestimate BWC for cows with a large BW loss (Figure 7). However, prediction of an absolute value of BWC might be less relevant for dairy producers if cows with a high BW loss can be correctly classified as such. Therefore, future applications should focus on classifying cows with a high versus a low BWC rather than predicting an absolute value for BWC.

Evaluation with previously unseen BWC records diminished the prediction performance model (Table 2). It is assumed that the model performance after cross-validation might be over-optimistic, due to the expected collinearity among FA. This limits the applicability of this model in the current state. Moreover, although data were randomly assigned to either the training or the evaluation set, the observed range in the training set was somewhat smaller than that in the evaluation set (see Table A1 in the Appendix A).

Other limitations to the model likely originate from underlying milk FA and BW records used for the prediction model. The milk FA was estimated from FTIR measurements with partial least squares algorithms that have a different degree of accuracy [31]. The coefficient of variation ranged from 3.6 to 12.8% for the main milk FA, with SFA and SCFA algorithms performing best with a coefficient of variation below 5%. The algorithms for the minor milk FA were weaker, with a coefficient of variation of 14.4% for PUFA and 27.2% for TFA. Milk FA measured by a reference method, such as gas chromatography, is expected to be more reliable for BWC predictions, but is not routinely used for milk recording samples, due to the high analytical costs and low analytical throughput. 

The prediction performance may also suffer from the daily fluctuations of BW measurements from floor scales, as previously observed [26], in particular in commercial dairy farms. These fluctuations commonly occur, due to daily variations in milk yield, feed intake, water intake, and time of milking in relation to the time of BW measurements, and might be thereby confounded with changes of physiological significance. In addition, due to the nature of the study using a large set of farm records, it was not possible to ascertain whether the floor scales were regularly calibrated.

The overall prediction performance might be further improved by a more frequent milk sampling scheme than the conventional once-a-month sampling by feeding the model with more relevant cow-side information during early lactation. This would better intercept the dynamics of milk composition change, due to changes in the metabolic status of the cow occurring after the onset of lactation. 

Further improvements to the prediction performance can be expected by including additional biomarkers in milk, such as acetone and citrate, that can be measured through FTIR and were previously associated with ketosis and negative energy balance in early lactation cows, respectively [32]. Future work should likewise focus on the use of the FTIR milk spectrum that may include additional model features to the milk composition and milk FA profile currently available. Despite the lack of a direct biochemical association between the FTIR spectrum and bodyweight change, as opposed to milk FA that peak in various regions of the FTIR spectrum, the potential of using FTIR spectra for physiological processes was previously shown, among others, for predicting the energy status of cows in early lactation [33,34]. 

## 4. Conclusions

The results of this study suggest that BWC might be estimated by FTIR milk FA profiles in milk recording samples. Nonetheless, before this prediction model can be used in commercial dairy farms, the model needs to be validated for the different country- or region-specific conditions, breeds, and herd management and feeding strategies. Further work analyses are needed to assess the impact of the level of BWC on milk production, reproductive performance, and health. For use in a real-time decision-support tool, predictions of BWC using early lactation test day may not be practical as any suggested modification in management strategies will likely not improve reproductive performance in the current lactation. However, the concept described here might help identify deficiencies on the herd level and improve overall herd management as a preventive strategy to improve herd performance in future lactations.

## Figures and Tables

**Figure 1 animals-10-01958-f001:**
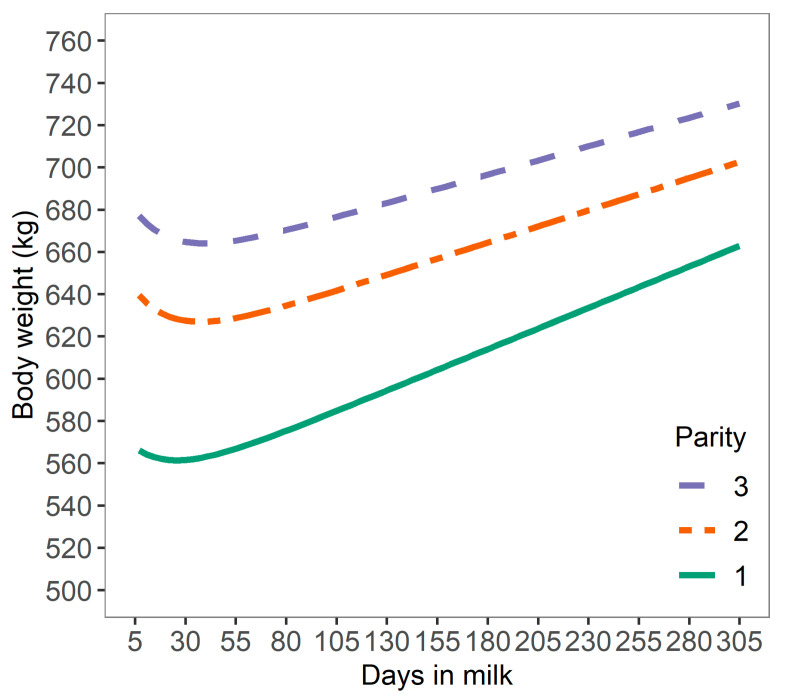
Daily smoothed bodyweight of dairy cows across 165 Danish Holstein herds from 7 to 305 days in milk.

**Figure 2 animals-10-01958-f002:**
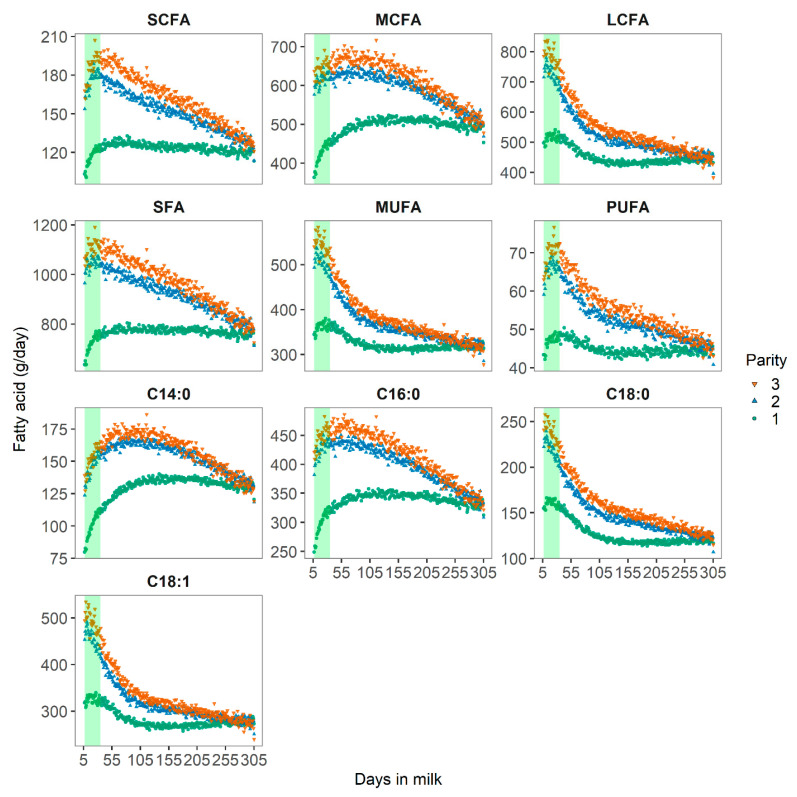
Distribution of short-chain (SCFA), medium-chain (MCFA), and long-chain (LCFA) fatty acids in g/day of Danish Holstein cows across 165 herds. The shaded vertical bars represent early lactation from DIM 7 to 35.

**Figure 3 animals-10-01958-f003:**
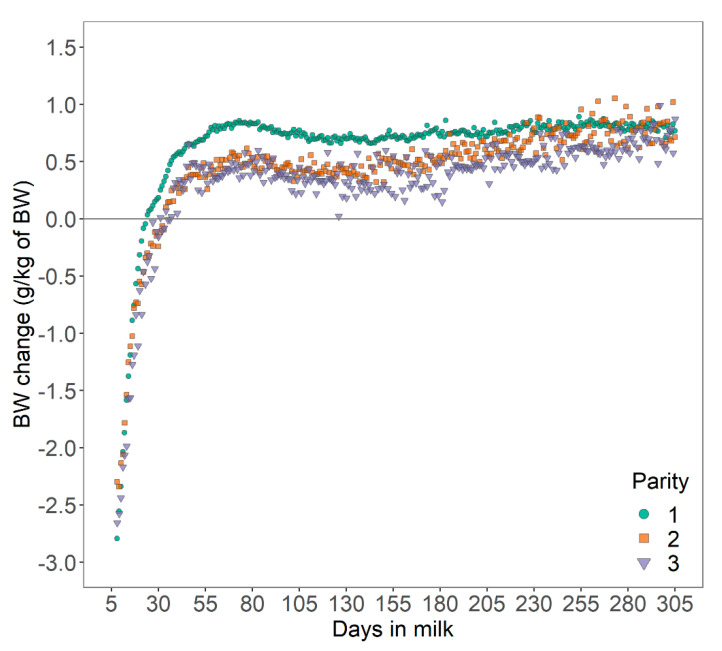
Mean daily bodyweight (BW) change of dairy cows across 165 Danish Holstein herds.

**Figure 4 animals-10-01958-f004:**
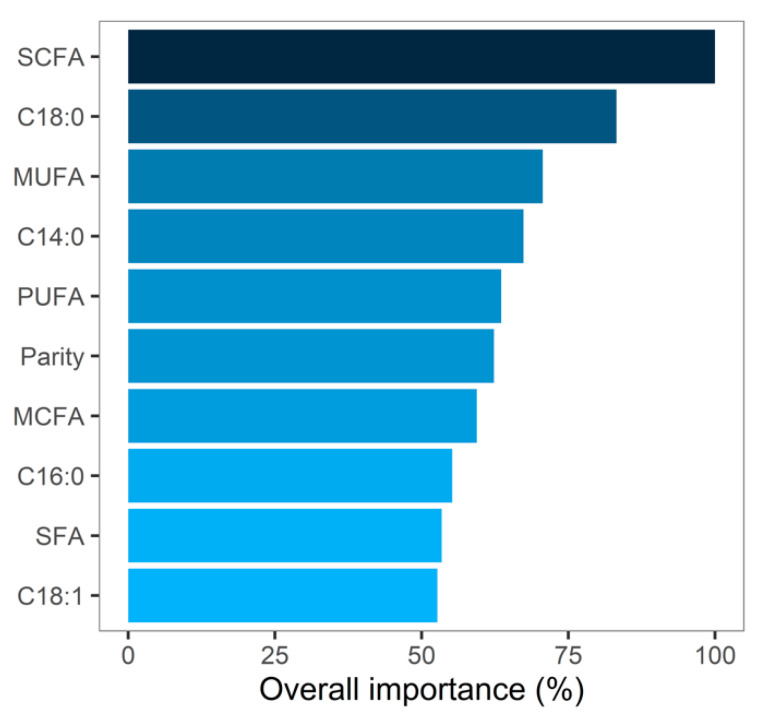
Relative overall importance after cross-validation of the top 10 predictors with the highest feature importance used by the random forest algorithm.

**Figure 5 animals-10-01958-f005:**
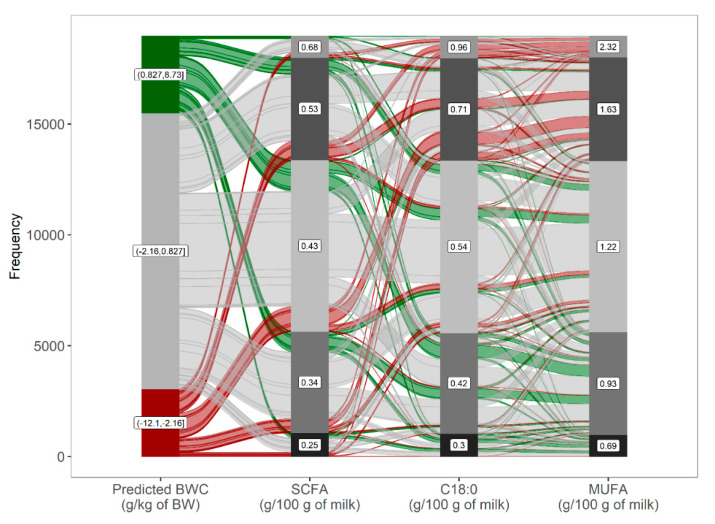
Model response, as shown by an alluvial diagram with predicted bodyweight change (BWC), ranked from negative to positive BWC as affected by the predictors with the highest feature importance. Nodes on the left denote categories of predicted BWC with the respective range for groups of cows in the bottom 10% (bottom node), intermediate (10–90%), and upper 10% (upper node) predicted BWC; nodes for predictors with respective means were fixed to five nodes based on the frequency distribution.

**Figure 6 animals-10-01958-f006:**
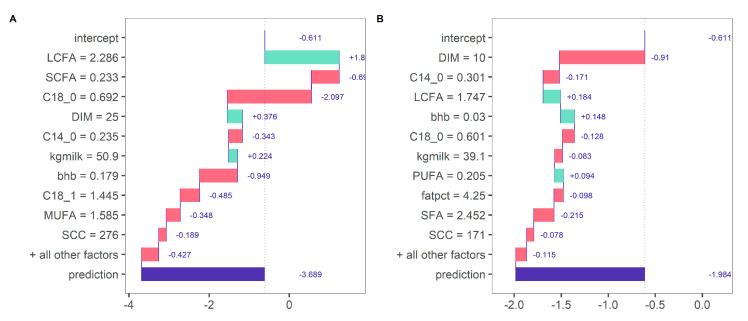
Contribution of variables for predicted bodyweight change (g/kg of BW per day; x-axis) of two randomly selected cows within the same herd experiencing a large BW loss during early lactation. Cow 36362 (DIM 25): −4.05 g/kg of BW per day (observed) versus −3.69 g/kg of BW per day (predicted) (**A**); cow 36357 (DIM 10): −2.22 g kg of BW per day (observed) versus −1.98 g/kg of BW per day (predicted) (**B**).

**Figure 7 animals-10-01958-f007:**
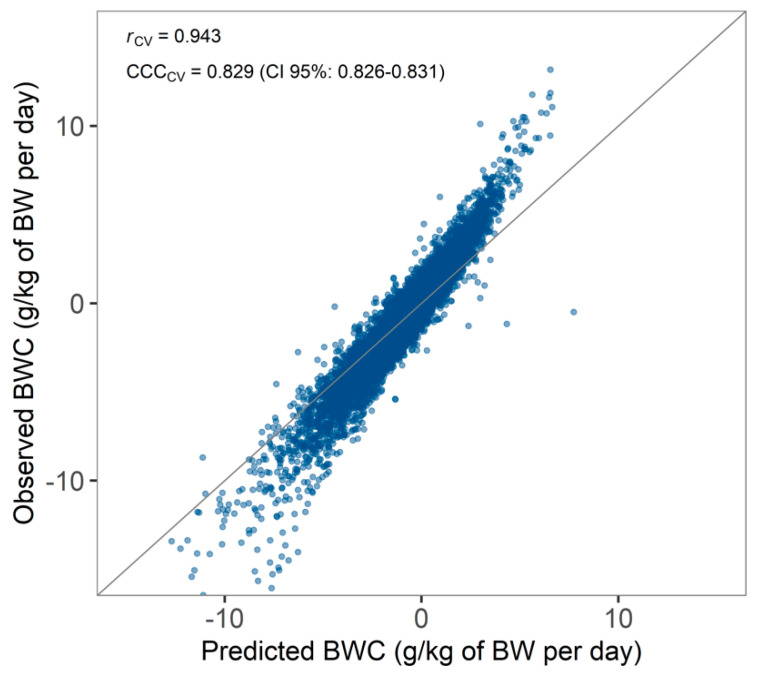
Predicted versus observed bodyweight change (BWC) with Pearson’s r and Lin’s concordance correlation coefficient (CCC) after cross-validation.

**Table 1 animals-10-01958-t001:** Test day records and bodyweights (BW) of 16,847 Danish Holstein cows in early lactation (7‒35 DIM) across 165 herds.

Trait	Parity 1 (*n* = 8113) ^1^				Parity 2 (*n* = 6698) ^1^				Parity 3 (*n* = 4327) ^1^			
	Mean	SD	P1	P99	Mean	SD	P1	P99	Mean	SD	P1	P99
Milk yield ^1^ (kg)	27.4	6.5	11.6	42.9	41.1	8.8	17.1	60.6	43.3	9.6	16.7	63.3
Fat ^1^ (%)	4.50	0.97	2.56	7.44	4.25	0.91	2.43	6.86	4.33	0.98	2.41	7.25
Protein ^1^ (%)	3.41	0.31	2.78	4.22	3.37	0.33	2.72	4.28	3.34	0.34	2.71	4.26
Ratio fat-to-protein ^1^	1.32	0.28	0.77	2.19	1.26	0.27	0.73	2.10	1.30	0.29	0.75	2.28
BHB ^1^ (mmol/L)	0.048	0.051	0.000	0.230	0.067	0.066	0.000	0.293	0.077	0.075	0.000	0.337
Fatty acid ^1^ (g/100 g of milk)												
C14:0	0.37	0.09	0.20	0.63	0.36	0.09	0.19	0.59	0.36	0.09	0.18	0.61
C16:0	1.11	0.25	0.62	1.87	1.05	0.25	0.56	1.74	1.05	0.26	0.56	1.81
C18:0	0.59	0.17	0.29	1.09	0.54	0.15	0.25	0.98	0.56	0.17	0.26	1.08
C18:1	1.20	0.40	0.54	2.46	1.11	0.35	0.51	2.22	1.15	0.40	0.51	2.48
SFA	2.68	0.58	1.48	4.39	2.57	0.57	1.33	4.03	2.60	0.60	1.40	4.32
MUFA	1.33	0.41	0.66	2.67	1.23	0.37	0.62	2.42	1.28	0.42	0.62	2.65
PUFA	0.17	0.05	0.08	0.31	0.16	0.04	0.01	0.28	0.16	0.04	0.07	0.30
SCFA	0.43	0.11	0.23	0.74	0.43	0.10	0.22	0.69	0.44	0.10	0.23	0.72
MCFA	1.59	0.37	0.84	2.68	1.52	0.38	0.75	2.51	1.52	0.39	0.76	2.64
LCFA	1.90	0.59	0.85	3.74	1.76	0.54	0.78	3.42	1.83	0.60	0.79	3.77
Initial BW ^2^ (kg)	572	60.3	432	717	641	64.8	491	804	681	70.7	523	854
Average BW ^3^ (kg)	603	66.1	458	772	656	67.7	505	834	689	70.2	531	870
BWC ^4^ (g/kg of BW per day)	−0.52	2.65	−7.77	5.17	−0.64	2.82	−9.06	5.59	−0.82	5.53	−10.50	8.57

Abbreviations used: SD = standard deviation, P1 = 1st percentile, P99 = 99th percentile. ^1^ Test-day records (abbreviations used: BHB = beta-hydroxybutyrate, SFA = saturated fatty acids, MUFA = mono-unsaturated fatty acid, PUFA = poly-unsaturated fatty acids, SCFA = short-chain fatty acid, MCFA = medium-chain fatty acid, LCFA = long-chain fatty acid); ^2^ At day in milk 7; ^3^ Between day in milk 7 and 35; ^4^ Bodyweight change.

**Table 2 animals-10-01958-t002:** Performance of a random forest algorithm to predict bodyweight change.

Model Metric ^1^	Grouped by Animal ^2^	Evaluation	Grouped by Herd ^3^	Evaluation
	Cross-Validation		Cross-Validation	
Pearson’s *r*	0.95	0.49	0.94	0.55
Lin’s CCC	0.85	0.39	0.83	0.46
R^2^	0.91	0.24	0.89	0.31
RMSEP	1.69	2.57	1.79	2.33
MAE	0.76	1.71	0.73	1.71
RPD	2.18	1.15	2.08	1.20
RPIQ	1.81	1.19	1.70	1.35

^1^ CCC = concordance correlation coefficient; RMSEP = root mean square error of the prediction; MAE = mean absolute error; RPD = ratio of performance to deviation; RPIQ = ratio of performance to the interquartile range. ^2^ Data splitting for model training (80%) and model validation (20%) stratified by cow. ^3^ Data splitting for model training (80%) and model validation (20%) stratified by the herd.

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
