# Peer review of "Fatty Acid Profiles from Routine Milk Recording as a Decision Tool for Body Weight Change of Dairy Cows after Calving"

_animals, 2020, doi:10.3390/ani10111958_

Round 1

Author Response

We thank the anonymous reviewer for the comments and suggestions. Please find attached our response to the comments below with reference to the specific changes made to the manuscript.

RESPONE TO REVIEWER 1

The manuscript reports the results of a study on the possibility to use the FA profile for predicting body weight change in the early lactation in cows. The study is well planned and organized, and the results are clearly presented.

So, I only have few comments:

The FA s content was predicted by FTIR spectroscopy, but the prediction has in general a low to medium accuracy (as the Authors themselves say at lines 286-87). It would be interesting to report some data on that, because the data on BWC are prediction from a prediction.

AU: Thank you for this comment, indeed the relative accuracy of the fatty acid prediction algorithms differ. Those for the minor milk fatty acids (poly-unsaturated and trans fatty acids) are the ones with the largest coefficient of variation.  We have added values (the coefficient of variation) for the main and the minor milk fatty acids (see lines 299-303).

Linked to this, another comment arises: in my opinion the paper would be more complete if the Authors included what they list as future work (lines 299-300), that is the use the FTIR spectrum instead of FAs prediction. It would be interesting to compare the results obtained with the two methods.

AU: Indeed, as mentioned in the original manuscript, we agree that using entire FTIR spectrum might help improving the accuracy of our model. However, while DHI agencies start appreciating the potential of FTIR spectra and start collecting them in central databases, they were not available for the milk samples used in the present study.
In addition, although several research teams, most prominently those involved in the European OptiMir project, showed the potential of using the entire milk spectrum, we can only hypothesize that this might work for body weight change predictions as well as there is no direct biochemical association between the milk spectra and body weight change as opposed to individual milk components that typically peak in various regions of the milk spectra. An amendment was insert in lines 323-327 with reference to previous published work (see chapter References).  

Reviewer 2 Report

Really enjoyed reading this paper. I have only some minor questions and comments.

1. Could you please state in the methods the time scale for each measurement? I could not find the information until the discussion. Test days were 30 d apart? body weight were daily or collected twice / 3 times a day at milking? does that mean that all measurements except for body weight change were 30 d apart?

2. Could you please comment on the RMSE vs the error associated with this type of scale (10%/load cell)? see lines 205-210

3.I was also curious about figure 1 vs the other weight figures. Figure 1 shows negative energy balance being over by 30 DIM. Fugure 3 shows it lasting past 30 days although it is hard to quantify from the graphs. I would expect negative energy balance to last longer than 30 d. Could the fact that you are collecting milk info only every 30 d mean that you are missing some of the dynamics of milk composition change and that is contributing to poor model predictability?

4. You used bodyweightchanges 3 d before  milk test to associate body weight change with milk parameters. Did you check to see if average body weight changes over a longer time scale than 3 d might have produced better predictions?

5. Please state in methods if scales were calibrated regularly and how.

6. Please add a footnote in tables 1 and A1 that defines P1 and P99.

7. Could the lack of predictability of LCFA be due to supplementation of rumen insoluble LCFA such as Megalac? Did you examine any diet information from these herds? Although mean milk fats are fairly high, was there any incidence of milk fat depression that could have changed milk FA profiles?

8. lines 199 and 201-please inlcude the words BW loss in the () with %. (I am assuming this is correct?)

9. line 225 - there is generally a lag in milk BHB and incidence of ketosis and with most ketosis showing up at 5-14 DIM, you may have missed ketosis since milk was only tested on testdays (30 d apart?). If you only examined cows with elevated BHB, would there be a relationship between LCFA or SCFA and BHB?

Author Response

We thank the anonymous reviewer for the comments and suggestions. Please find attached our response to the comments below with reference to the specific changes made to the manuscript.

RESPONSE TO REVIEWER 2

Really enjoyed reading this paper. I have only some minor questions and comments.

  1. Could you please state in the methods the time scale for each measurement? I could not find the information until the discussion. Test days were 30 d apart? body weight were daily or collected twice / 3 times a day at milking? does that mean that all measurements except for body weight change were 30 d apart?

AU: Thanks for pointing this out, an existing explanation was moved to a more prominent position (Line 84-85) and additional amendments were made in the method section (Line 85-87). Test days records originated from generally monthly test days but only records between 5 and 35 day in milk were retained for analysis which means that for most cows (87%) only one test day was retained, and the remainder had mostly 2 test days. In contrast, body weights were recorded at each milking, thus each time the cow visited the milking robot as the scales were installed at the bottom of the milking robot (included in the original manuscript in Line 89-91).

  1. Could you please comment on the RMSE vs the error associated with this type of scale (10%/load cell)? see lines 205-210

AU: We have made an amendment (Line 217-222) and discussed the effect of the 10% error of this type of floor scales with regard to the body condition score attained.

3.I was also curious about figure 1 vs the other weight figures. Figure 1 shows negative energy balance being over by 30 DIM. Fugure 3 shows it lasting past 30 days although it is hard to quantify from the graphs. I would expect negative energy balance to last longer than 30 d. Could the fact that you are collecting milk info only every 30 d mean that you are missing some of the dynamics of milk composition change and that is contributing to poor model predictability?

AU: Indeed, test days were generally available only every 30 days. It is common practice to test once a month, however a more frequent milk sampling scheme during the early lactation would be definitely best to catch the dynamics of milk composition change due to changes in the metabolic status of the cow. A statement was included in the original manuscript but rephrased in line with this comment (Lines 315-317).

  1. You used bodyweightchanges 3 d before  milk test to associate body weight change with milk parameters. Did you check to see if average body weight changes over a longer time scale than 3 d might have produced better predictions?

AU: It was pointed out in literature that changes in milk fatty acids due to changes in feed rations occur within mostly 2 days and up to 4 days (Elgersma et al., 2004; https://doi.org/10.1016/j.anifeedsci.2004.08.003). We have decided to opt for a 3-day average to catch any major change in the milk fatty acid profile, but as no literature findings suggested that changes would occur after a longer timer period, we have not evaluated potential differences between a 3-day average or over more than 3 days. As we missed to include the reference in the original manuscript, an amendment was made to Lines 114-116.

  1. Please state in methods if scales were calibrated regularly and how.

AU: Amendments were added in the method section (Lines 92-95) and discussion section (Lines 311-312). Due to the particularities of this study using farm records, it was not possible to ascertain if the participating producers followed the recommended calibration procedures.

  1. Please add a footnote in tables 1 and A1 that defines P1 and P99.

AU: Footnotes added in Table 1 and Table A1.

  1. Could the lack of predictability of LCFA be due to supplementation of rumen insoluble LCFA such as Megalac? Did you examine any diet information from these herds? Although mean milk fats are fairly high, was there any incidence of milk fat depression that could have changed milk FA profiles?

AU: Unfortunately, with the current approach based on a large dataset of farm records, diet information were not integrated with test day information and, thus, not available. As a matter of fact, we are currently working with a smaller set of approximately 100 dairy herds for which diet information were collected at each test day, with the aim to associate changes in fatty acids levels to dietary changes and supplementations with rumen insoluble LCFA. An amendment was made to in line with this comment (Lines 262-263).

  1. lines 199 and 201-please inlcude the words BW loss in the () with %. (I am assuming this is correct?)

AU: Correct (Lines 211 and 213)

  1. line 225 - there is generally a lag in milk BHB and incidence of ketosis and with most ketosis showing up at 5-14 DIM, you may have missed ketosis since milk was only tested on testdays (30 d apart?). If you only examined cows with elevated BHB, would there be a relationship between LCFA or SCFA and BHB?

AU: As a matter of fact, we have examined a possible relationship with BHB in recent data across over 70,000 first test day records in Eastern Canada and found that certain LCFA (in particular C18:1) was correlated to BHB but only to a certain extent (< 0.29). These findings further strengthened our conviction that milk fatty acid profiles can give an added value to existing tools such as milk BHB in alerting dairy producers towards possible metabolic disorders during early lactation.